# An Intelligent Model and Methodology for Predicting Length of Stay and Survival in a Critical Care Hospital Unit

Enrique Maldonado Belmonte *, Salvador Oton-Tortosa, Jose-Maria Gutierrez-Martinez and Ana Castillo-Martinez

Department of Computer Science, University of Alcalá, 28805 Alcalá de Henares, Spain;
salvador.oton@uah.es (S.O.-T.); josem.gutierrez@uah.es (J.-M.G.-M.); ana.castillo@uah.es (A.C.-M.)
* Correspondence: e.maldonado@edu.uah.es; Tel.: +34-918856645

**Abstract:** This paper describes the design and methodology for the development and validation of an intelligent model in the healthcare domain. The generated model relies on artificial intelligence techniques, aiming to predict the length of stay and survival rate of patients admitted to a critical care hospitalization unit with better results than predictive systems using scoring. The proposed methodology is based on the following stages: preliminary data analysis, analysis of the architecture and systems integration model, the big data model approach, information structure and process development, and the application of machine learning techniques. This investigation substantiates that automated machine learning models significantly surpass traditional prediction techniques for patient outcomes within critical care settings. Specifically, the machine learning-based model attained an F1 score of 0.351 for mortality forecast and 0.615 for length of stay, in contrast to the traditional scoring model's F1 scores of 0.112 for mortality and 0.412 for length of stay. These results strongly support the advantages of integrating advanced computational techniques in critical healthcare environments. It is also shown that the use of integration architectures allows for improving the quality of the information by providing a data repository large enough to generate intelligent models. From a clinical point of view, obtaining more accurate results in the estimation of the ICU stay and survival offers the possibility of expanding the uses of the model to the identification and prioritization of patients who are candidates for admission to the ICU, as well as the management of patients with specific conditions.

**Keywords:** big data; machine learning; architecture; interoperability; computing methodologies

## 1. Introduction

The use of information systems in clinical patient care has recently grown, generating a high volume of data, as confirmed by Danciu et al. [1]. These data come from different sources, such as clinical reports, diagnostic test results, monitoring devices, surveys on clinical processes or the care received, and almost any type of documentation related to a medical act. This information is heterogeneous, ranging from numerical quantification to clinical observations in natural language. As a result, the volume of data is very high and requires the use of technologies that allow its effective exploitation. Potential applications of technology in healthcare include system integration, big data, and artificial intelligence [2]. The use of artificial intelligence, more specifically the areas of machine learning and deep learning, offers the opportunity to obtain information and draw conclusions from large amounts of data through processes that resemble the way human knowledge and reasoning are constructed [3]. The volume of this information, together with the high level of detail it provides, contributes to overcoming the barriers and limits of traditional analyses [4]; technology not only allows information to be analyzed more efficiently than using human resources, but it also permits a more in-depth analysis that allows conclusions to be drawn that would otherwise be impossible [5].

Among the ways data analysis can provide knowledge that improves multiple areas of the health sector, one of the most important is predicting the evolution of a patient's clinical condition [6,7], i.e., estimating a patient's condition over time.

Estimating patient progression in critical care, where the number of patients that can be treated is very limited and the cost of care per patient is high, could be a relevant achievement in this type of estimation. On the other hand, the criteria for admission to these units are subjective to a high degree, despite the effort to standardize them, as Kim et al. concluded [8], and this could also be assisted by AI. Based on the relevant clinical data of the patients, it could objectively estimate the probability of survival and the estimated length of stay in the critical care unit. Models like this have been developed using machine learning techniques, which are capable of establishing patterns and drawing conclusions by processing a large volume of historical data. A model with these characteristics could potentially enhance safety in decision-making processes, thereby helping to reduce subjectivity, always as a form of support for clinicians in decision making.

However, the greatest weakness of these systems lies in the interpretation of the information obtained. The predictive models obtained using machine learning techniques are considered a black box because, in practice, it is impossible to determine with reasonable effort the rationale for a given prediction [9]; this is because it is based on the analysis of multiple parameters from thousands of previous records. Thus, in these models, there is no certainty that the conclusions it provides are reliable since there is no traceability justifying the conclusion, nor is it possible to carry out a strict demonstration of the goodness of the model [10]. Therefore, validation of the model is essential, since some of the responsibility for these important decisions is assumed by the doctors in the system. Due to this, it is essential to apply a methodology that establishes the steps to follow in the development of an intelligent model, validation against a traditional model, and establishment of the acceptable thresholds of deviation in the results. Similar analysis methodologies have been proposed and successfully applied in virtual screening [11].

Another aspect to be considered is the effectiveness of these AI-based systems. If we compare their results with linear regression, considered as traditional here, we will obtain results that are inferior [12]. However, reviews of the literature indicate that the median sample size used is 1250 records [13]. The use of a larger sample size is expected to provide better results that justify the use of machine learning to make predictions, as has been demonstrated in specific predictions concerning critically ill patients with greater volumes of data [14]. The possibility of using machine learning to obtain better results than scoring systems in the healthcare field has been recognized [15]. It has been used in specific applications such as the prediction of upper gastrointestinal bleeding [16], so we believe it is feasible to investigate it for predicting the evolution of critical patients.

In the following sections of the paper, we detail the objectives to be covered, describe the methodology used, analyze the results, and present the conclusions.

## 2. Objectives

The main objective of the present work is to forecast the stay time and survival chances of patients in an intensive care unit with a higher degree of reliability than that obtained by current systems.

To this end, the creation of an intelligent model was proposed with the necessary methodology to ensure its validity and ability to forecast the length of stay and survival rate in the critical hospitalization area with greater reliability than current practices.

The following complementary objectives are proposed, derived from the use of artificial intelligence in assessing critical patients:

- To compare the goodness of the results obtained compared to a traditional scoring system, which will confirm the validity of the generated model and allow its use with a sufficient degree of confidence.
- To propose and apply a methodology for the creation of both models and their comparison, and establishing the parameters of validity of the results.

## 3. Materials and Methods

In this section, we first introduce the dataset. The research is based on a comprehensive dataset that has been collected and maintained over several years in a clinical setting. Although these data were not initially gathered with a specific research objective in mind, their depth and breadth provide a remarkable opportunity for developing advanced predictive models that can significantly impact the care of critical patients.

Once the data set is presented and the learned lessons about the requirements of processing are described, a methodology is proposed for creating an intelligent model with a high grade of maintainability and repeatability in its tasks, which use higher amounts of data than previous models and try to obtain better results than traditional models.

The generated model, if the metrics obtained are considered to be sufficiently satisfactory, could be used as support for decision making by clinical professionals.

### 3.1. Data Set

This study uses the historical data from critical patients obtained from a reference clinical center. The data set used in this study covers the totality of patients treated in the critical care unit of a major clinical center located in a large urban area. This large data set includes a diverse patient population representative of the city's broad demographics.

Specifically, data related to the clinical information of patients who were hospitalized in the critical care unit from the time prior to admission until discharge were used. For this study, historical data were available for episodes occurring between 1 January 1999 and 31 December 2023. The data were collected from patients located in the geographical area of Madrid, where the clinical center is situated.

The data were subjected to a comprehensive analysis followed by an exhaustive cleansing process, which was essential to ensure the integrity and accuracy of the dataset. This included a thorough verification of the fields, which was critical to confirm both the completeness and consistency of the processing. Records failing to meet these criteria were excluded from the analysis. Consequently, this rigorous approach yielded a refined dataset comprising 24,876 episodes of critical hospitalizations, each characterized by complete and consistent data suitable for constructing predictive models.

The patient population was categorized by sex and age as follows: 14,141 males, 9392 females, and 1343 unspecified. Regarding age, 316 patients were under 20 years; 3359 patients were aged 20 to 40 years; 7567 patients fell into the 40 to 60 years range; 11,416 patients were between 60 and 80 years; and 2218 patients were over 80 years of age.

Predictive variables were obtained based on the synthetic criteria defined by the Philip R. Lee Institute for Health Policy Studies (PRLI criteria) based on Lombardozi's assessments [17]. These criteria reflected the clinical situations of patients who were candidates for ICU admission, allowing for an assessment of their health status.

### 3.2. Methodology

With this data set and the identified outcome variables, length of stay and survival rate, we set up a method to ensure the development of a machine learning model trained with the predictive variables and verified it against traditional algorithmic analyses. Although the starting point included a data set with information from many years ago, the methodology included the processing and validation of integrated historical and current data.

This methodology also ensures the repeatability of the process, maintaining the quality of the results and, if necessary, supporting the creation of new models in the field. The methodology consists of a set of four interconnected phases divided into a list of tasks, as shown in Figure 1.

The four phases of the methodology and their associated tasks are presented in the following subsections.

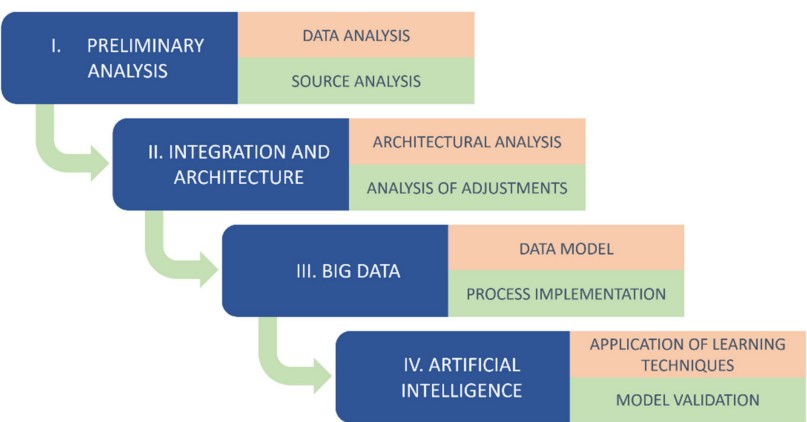

**Figure 1.** Methodological phases of the development of an intelligent model for length of stay and survival in a Critical Care Hospital Unit.

### 3.2.1. Preliminary Analysis

This phase focuses on the identification of the clinical criteria that allow for the predictions sought in the research. A compilation of information requirements and a list of data sources to be used are obtained.

To achieve this, the set of clinical data that is representative of the patient's situation, is available, and can be obtained in an automated way from heterogeneous data sources must be identified. Ensuring representativeness is the focus of this phase, while the availability of information is considered in the next phase, although both data characteristics are closely related.

It is necessary to list all available sources of information, as well as their characteristics and details regarding format, periodicity, and historical and future availability.

In the specified case study, a systematic identification of pertinent clinical criteria essential for the predictive models was conducted. This process included an extensive compilation of requisite information and the identification of relevant data sources that were utilized throughout the study.

### 3.2.2. Integration and Architecture

One of the main tasks is the creation of an architecture that allows access to clinical data and ensures their compatibility with the automation of the process. To achieve this goal, existing architectures are analyzed, and the necessary adaptations and extensions are proposed. In this phase, an architecture model adapted to the data processing needs is obtained.

It is required to work with the existing technical architectures that manage the information selected in the previous phase. It is necessary to analyze the capacity of each information source to provide the necessary information, as well as the data storage in terms of format and sampling frequency, and to determine whether the information can be used by the model, automatically feeding the process.

Design and development of the adaptations or adjustments is necessary to meet the needs that current architecture does not address. It is also necessary to establish the destination of the data to structure and prepare it for use in the model.

In the context of the current case study, the existing data architecture was carefully adapted to facilitate access to and compatibility with clinical data, essential for automating the research processes. This adaptation targeted the comprehensive data processing demands of the developed model, ensuring that all architectural components were suitably aligned to support the investigative objectives.

### 3.2.3. Big Data

The efficient use of information requires a homogeneous repository to ensure the availability of up-to-date data. This requires the definition of a data model capable of

providing enough information for machine learning systems. In addition, it must be able to perform information retrieval processes to populate the model. In this phase, the information model is defined and the necessary data are loaded.

A data model must be created that can receive all the information from the selected sources in the necessary formats, homogenize it, and convert it into a unified model that can be used as an input by the machine learning tools.

To populate the data model with information, it is necessary to implement extraction, transformation, and loading (ETL) mechanisms for each information source. All processes are designed and tested, establishing the tools to be used as well as the set of information flows proposed with the different ETLs. The execution of the ETLs generates a loaded model ready to be used by the machine learning system. The model continues to receive information through the periodic ETLs that were defined, allowing for the evolution and adjustment of the intelligent model to the new data received.

For the case study under consideration, specific technologies were thoughtfully chosen. The design and implementation of robust ETLs were carried out to populate the information model. This model was structured to ensure the continuous availability of updated data, which is crucial for the functioning of machine learning systems.

### 3.2.4. Artificial Intelligence

Once all the data needed for the model were available, it was necessary to characterize the outcome variables and apply the required techniques. The goal of the survival measurement was clear: the outcome would be either positive or negative. However, the length of stay in the ICU can range from a few hours to more than a year. Therefore, the time periods were separated into two differentiated blocks: short stays (less than or equal to 3 days) and long stays (greater than 3 days). This division is useful from a clinical perspective for the estimation of the use of resources of the critical care unit.

Appropriate techniques were then applied to generate several models for both outcome variables, and the one that provided the most accurate result was selected. In this phase, an intelligent predictive model was obtained.

A machine learning process was carried out to obtain the predictive model that provided the greatest accuracy. To do this, tests were carried out with different artificial intelligence algorithms to identify the one that correctly classified an element.

The primary objective was to demonstrate significant enhancements over traditional scoring systems, rather than achieving the definitive optimization of the model. For this goal, tuning techniques such as hyperparameter optimization and model stacking were employed to enhance model performance. While it was not intended to exhaustively fine-tune each parameter, the application of these methods was essential to ensure that the machine learning models were accurate enough to represent a practical and meaningful improvement.

The process was carried out as follows: A first approximation was made using various machine learning techniques. Optimizations were then performed on each of these models, obtaining results that allowed us to discern those with the best metrics. Among these models, a stacking model was proposed to obtain the most solid model for the prediction of both outcome variables. The stacking model was an ensemble learning technique. This approach combined the predictions of multiple base machine learning models using a meta-classifier. The base models were trained in parallel, and the meta-classifier learned to optimally combine their predictions to improve the final predictive accuracy and generalizability over any single model.

The results obtained with the best selection of techniques were analyzed according to the established quality factors. The following factors were selected for assessment, accuracy, precision, and recall [18], as well as the synthetic indicator f-beta, which depends on precision and recall.

- Accuracy reflects the percentage of correctly classified items. This is a clear indicator in the case of balanced data sets. If there is a majority class, it will be less so, as models tend to classify toward it.
- Precision denotes the percentage of reliability of the predictions made for the membership of elements in a class. That is, out of the total number of elements that are estimated to belong to a class, we determine how many actually belong to it. This value allows us to assess the model's capability by identifying the generation of false positives.
- Recall shows the percentage of reliability for identifying whether an item belongs to a class, that is, out of the total number of elements belonging to a class, it shows the number that are correctly predicted. With this value, it is possible to assess the capability of the model by identifying the generation of false negatives.
- F-beta combines precision and recall from the harmonic mean of both. This value provides an objective assessment in terms of accuracy and completeness.
- ROC AUC is a metric assessing a classification model's ability to distinguish between classes; higher values indicate superior performance.

In the case study under review, a series of advanced artificial intelligence techniques was utilized to develop predictive models. This application encompassed the selection of appropriate AI methodologies and their strategic implementation to achieve optimal prediction accuracy.

In addition, a comparative analysis was carried out with a reference method, the scoring system based on the PRLI criteria.

## 4. Results

In this section, we report on the different tasks involved in the proposed methodology, providing details of its application to the historical and current data.

### 4.1. Preliminary Analysis

This phase contained two tasks that analyzed the data and data sources.

Concerning the data, we proposed using the PRLI criteria, which have been used in several previous studies that considered data from multiple ICUs as a reference for the assessment of critically ill patients [19,20]. These criteria are shown in Table 1.

**Table 1.** Description and typology of the Philip R. Lee Institute criteria for evaluating patients who are candidates for admission to the ICU.

| Typology | Description |
|---|---|
| Physiological parameters | Deep coma<br>Heart rate greater than 150 beats per minute<br>Systolic pressure below 90 |
| Chronic illnesses | Chronic kidney failure<br>Positive diagnosis of cirrhosis<br>Positive diagnosis of cancer |
| Acute pathology | Acute kidney failure<br>Severe arrhythmia<br>Stroke incident<br>Severe gastrointestinal bleeding<br>Intracranial lesion with mass effect |
| Additional parameters | Age 65 or over<br>Age 84 or over<br>Cardiac and pulmonary resuscitation needed<br>Need for mechanical ventilation<br>Emergency admission |

Previous studies were carried out by the authors using these criteria with similar objectives, although using traditional calculation techniques and obtaining results that need to be improved in this study [21]. The selected criteria indicated in Table 1 represent the relevant information for the assessment of patients using a reduced number of values, all of them of binary type and obtained in an automated way from heterogeneous data sources. This last characteristic is what makes them suitable for use in the application of an intelligent model that needs a large amount of data to be formalized.

Concerning the sources, to obtain the PRLI criteria for patients, the main source of information was the electronic medical record. Although historical information about stays in the critical unit was available, the acquisition of the PRLI criteria was not straightforward because they are not routinely used, and in most cases, thus, it was necessary to use natural language processing techniques and an analysis of clinical episodes to delimit the values of the parameters and data on stays in the intensive care unit.

Moreover, it was necessary to retrieve information related to vital signs obtained from electromedical devices and to access the corresponding information repository systems linked to these devices.

Therefore, to develop the model, clinical information systems (HIS and departmental applications) as well as electromedical monitoring devices were considered as sources of information.

### 4.2. Integration and Architecture

To set up the architecture for this model's creation, we used the architecture elaborated upon in a previous work for the periodic retrieval and management of clinical data as a starting point [22]. Figure 2 shows this architecture created for the integration of different clinical systems in real time, allowing for consistency among data from all systems. However, it used direct integration with some information systems to retrieve all the necessary data. To achieve this, it employed an integration engine and supporting web services as an interoperability mechanism. For this integration, it adhered to interoperability standards (HL7 [23], DICOM [24], SNOMED-CT [25], ICD [26], LOINC [27]) that guaranteed the platform's scalability and evolution possibilities.

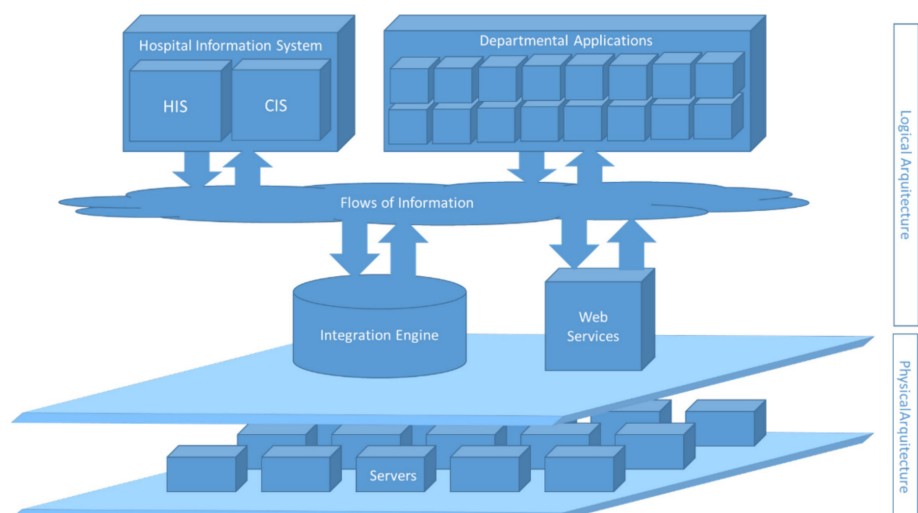

**Figure 2.** Architecture for information management that contemplates the integration between clinical information systems.

This solution allowed for the integration of the different systems of a clinical center, covering the operational needs of the center, both technical and functional. However, this pre-existing system was not oriented to the exploitation of information and data analysis from other systems. Therefore, it was necessary to make adaptations to use the clinical data stored as the source of the intelligent model.

Adjustments to the architecture tried to cover the needs of data exploitation and analysis. Massive repositories were incorporated to enable the storage of clinical information.

On the one hand, a data lake was implemented which stored all the structured clinical and textual information, as it was unnecessary to include images or videos from diagnostic tests. On the other hand, a data warehouse was created which maintained the records for subsequent analysis and exploitation.

To develop the model, considering that the sources of information were the clinical information systems (HIS and departmental applications) and the electromedical monitoring devices, an extension of the architecture was defined, incorporating the data lake and the data warehouse.

Operational interoperability remained unchanged with respect to the base architecture used—an integration engine coordinated the exchange of messaging between information systems—and web services were used as a support mechanism. The schematic of the resulting architecture is presented in Figure 3.

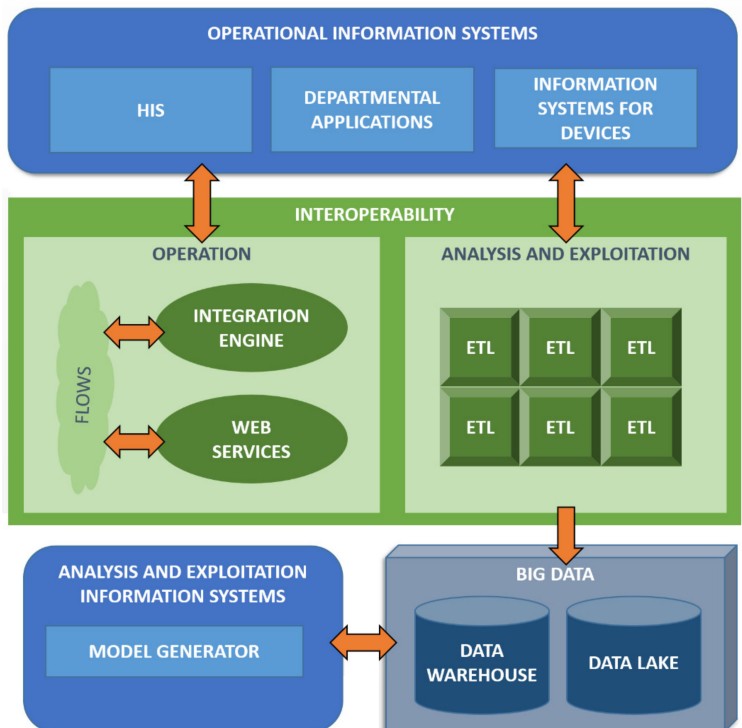

**Figure 3.** Diagram of the information architecture focused on data exploitation.

### 4.3. Big Data

This phase was concentrated on the integration of different data sources. To achieve this, a data model was developed with all the information required for the elaboration of the intelligent model, covering the parameters analyzed in the first phase. It was developed using specific tables from the data warehouse that contained the necessary clinical information about the physiological parameters, chronic diseases, acute pathologies, and additional parameters related to the clinical episodes of critical hospitalization.

In the implementation of the process, the following technologies were used to develop the ETLs:

- Protocols: the standard HL7 protocol was adopted for the exchange of clinical information between systems;
- Databases: Oracle SQL databases were used as repositories both for the clinical application tables and for the data lake and data warehouse;
- Periodic ETLs: the Pentaho Business Intelligence system was available for the development, planning, and execution of scheduled periodic ETLs;

- Custom development: Python was used for the development of ad hoc ETLs to exchange information and calculations;
- The circuits of the ETLs needed to feed the model were defined, identifying different categories;
- Periodic processes facilitated uploading from the HIS and electromedical devices to the data lake and critical patient system;
- Information retrieval process: Through analysis of clinical records from the HIS, the data lake, and the critical patient system, the relevant data for the development of the model were retrieved;
- Registration processes: The data obtained in the information retrieval processes were registered and stored in the data warehouse;
- The logical order of the ETLs followed the order of the blocks (1–6) in Figure 4.

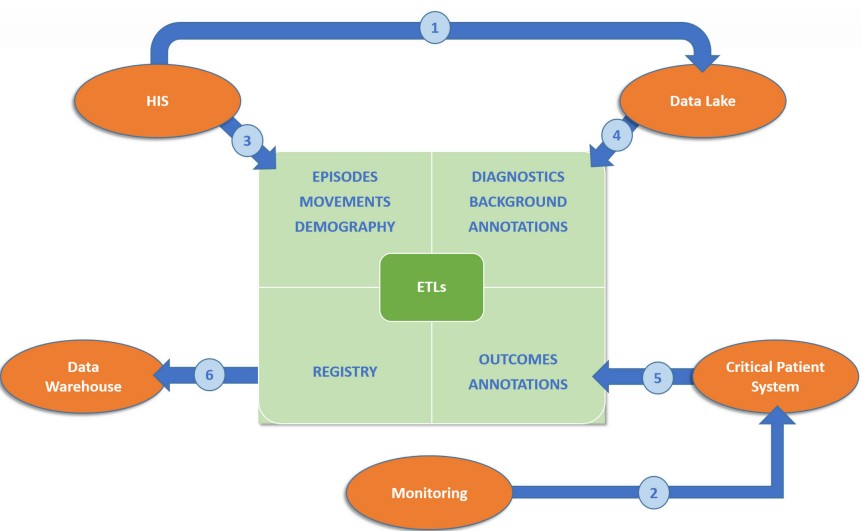

**Figure 4.** Information flows for data processing using the proposed model.

ETL processes were designed, implemented, and commissioned as indicated in Table 2.

**Table 2.** Specifications of the extraction, transformation, and loading processes (ETLs) for the proposed model.

| Category | Block | Destiny | Origin | Data |
|---|---|---|---|---|
| Periodical Processes | 1 | Data lake | HIS | Diagnostics |
| | 1 | Data lake | HIS | History |
| | 1 | Data lake | HIS | Notes |
| | 2 | Critics' system | Monitors | Heart Rate |
| | 2 | Critics' system | Monitors | Blood Pressure |
| Recovery Processes | 3 | - | HIS | Episodes |
| | 3 | - | HIS | Movements |
| | 3 | - | HIS | Demography |
| | 4 | - | Data lake | Diagnostics |
| | 4 | - | Data lake | History |
| | 4 | - | Data lake | Notes |
| | 5 | - | Critics' system | Heart Rate |
| | 5 | - | Critics' system | Blood Pressure |
| | 5 | - | Critics' system | Notes |
| | 5 | - | Critics' system | Heart Rate |
| | 5 | - | Critics' system | Blood Pressure |
| | 5 | - | Critics' system | Notes |
| Registration Processes | 6 | Data warehouse | - | All |

To obtain information from the retrieval process category, a natural language pattern identification system was used to identify specific clinical situations [28].

Once the development of the ETLs for the categories of retrieval and recording processes was complete, the supervised execution of the same was carried out to load the data model necessary to obtain the intelligent model.

Once all the information was available, records containing erroneous or inconsistent information were eliminated, finally obtaining a total of 24,876 critical hospitalization episodes with valid information.

### 4.4. Artificial Intelligence

This section contains the application of the AI techniques. Due to its complexity, we present it divided into subsections.

#### 4.4.1. Application of Learning Techniques

The learning process was conducted using the following technologies: Python 3.6.9, with the Anaconda Navigator 2.5.2 distribution and the machine learning libraries FastAI 0.7.0, scikit-learn 0.24.2, and imbalanced-learn 0.8.1.

Two learning processes were carried out to obtain two different models: one to predict patient survival during their stay in the ICU and the other to predict the length of stay in the ICU.

For the estimation of the length of stay, we defined the separation into two types: those of short stay (less than or equal to three days) and those of long stay (more than three days).

To obtain the model, an initial approach was made to the data using the following machine learning algorithms that can be used for classification [29]: Logistic Regression [30], Random Forest [31], Support Vector Machines [32], Gradient Boosting [33], Neural Network [34] and K-Nearest Neighbors [35].

The metrics were obtained and are indicated in Tables 3 and 4.

**Table 3.** Metrics of the machine learning algorithms for initial approximation of mortality.

| Techniques | Mortality | | | | |
|---|---|---|---|---|---|
| | ROC AUC | F1 Score | Precision | Recall | Accuracy |
| Logistic Regression | 0.640 | 0.0 | 0.0 | 0.0 | 0.832 |
| Random Forest | 0.623 | 0.041 | 0.162 | 0.024 | 0.816 |
| Support Vector Machines | 0.584 | 0.0 | 0.0 | 0.0 | 0.832 |
| Gradient Boosting | 0.649 | 0.002 | 0.5 | 0.001 | 0.832 |
| Neural Network | 0.647 | 0.009 | 0.190 | 0.004 | 0.830 |
| K-Nearest Neighbors | 0.591 | 0.144 | 0.243 | 0.102 | 0.797 |

**Table 4.** Metrics of the machine learning algorithms for initial approximation of length of stay.

| Techniques | Length of Stay | | | | |
|---|---|---|---|---|---|
| | ROC AUC | F1 Score | Precision | Recall | Accuracy |
| Logistic Regression | 0.580 | 0.674 | 0.574 | 0.816 | 0.572 |
| Random Forest | 0.633 | 0.726 | 0.561 | 0.726 | 0.543 |
| Support Vector Machines | 0.536 | 0.633 | 0.574 | 0.680 | 0.574 |
| Gradient Boosting | 0.584 | 0.678 | 0.574 | 0.827 | 0.573 |
| Neural Network | 0.577 | 0.650 | 0.579 | 0.740 | 0.567 |
| K-Nearest Neighbors | 0.531 | 0.570 | 0.557 | 0.584 | 0.522 |

Optimizations were performed on the different models, mainly using the class-balancing (SMOTE) and hyperparameter optimization (Grid Search) techniques.

Optimizations were then performed on the application of each of the machine learning techniques. The metrics were obtained and are indicated in Tables 5 and 6.

**Table 5.** Metrics of the optimized machine learning algorithms for mortality (class balancing and hyperparameter optimization techniques).

| Techniques | Mortality | | | | |
|---|---|---|---|---|---|
| | ROC AUC | F1 Score | Precision | Recall | Accuracy |
| Logistic Regression | 0.640 | 0.333 | 0.229 | 0.608 | 0.593 |
| Random Forest | 0.654 | 0.353 | 0.240 | 0.667 | 0.592 |
| Support Vector Machines | 0.656 | 0.347 | 0.230 | 0.709 | 0.554 |
| Gradient Boosting | 0.645 | 0.350 | 0.252 | 0.571 | 0.645 |
| Neural Network | 0.635 | 0.342 | 0.228 | 0.685 | 0.560 |
| K-Nearest Neighbors | 0.620 | 0.312 | 0.247 | 0.422 | 0.689 |

**Table 6.** Metrics of the optimized machine learning algorithms for length of stay (class balancing and hyperparameter optimization techniques).

| Techniques | Length of Stay | | | | |
|---|---|---|---|---|---|
| | ROC AUC | F1 Score | Precision | Recall | Accuracy |
| Logistic Regression | 0.580 | 0.629 | 0.577 | 0.693 | 0.558 |
| Random Forest | 0.584 | 0.607 | 0.589 | 0.627 | 0.560 |
| Support Vector Machines | 0.570 | 0.591 | 0.593 | 0.588 | 0.558 |
| Gradient Boosting | 0.575 | 0.607 | 0.581 | 0.636 | 0.554 |
| Neural Network | 0.578 | 0.581 | 0.595 | 0.569 | 0.556 |
| K-Nearest Neighbors | 0.548 | 0.572 | 0.576 | 0.569 | 0.539 |

When analyzing the results, it could be seen that with respect to mortality, the K-Nearest Neighbors algorithm had the highest accuracy but a low recall, indicating that it tended to be accurate, but with a low efficiency in identifying positive cases. Gradient Boosting, Random Forest, and Neural Network showed balanced performances. Support Vector Machines had high recall, minimizing false negatives, but with low precision. Logistic Regression was the least effective. With respect to the stay, Logistic Regression had the highest F1 score, with a good balance between precision and recall, although with unremarkable accuracy and ROC AUC. Random Forest and Gradient Boosting showed balanced results for all metrics. Neural Network had high precision but medium recall, and K-Nearest Neighbors has low accuracy, with no outstanding results in any metric.

We then implemented a staking model using Gradient Boosting and SVC as base models. We used Gradient Boosting for mortality because it had high accuracy, a relatively high ROC, and could distinguish effectively between patient types, while for length of stay, it presented a balanced performance with reasonable accuracy and F1 score values. We used SVC for mortality because it had high recall, which is critical to predict deaths correctly and minimize negative biases.

We used Logistic Regression as the meta-classifier because it is computationally simple and allows for easy interpretation of the results.

Once the stacking model was implemented, class balancing and hyperparameter optimization techniques were utilized. The metrics were obtained and are indicated in Table 7.

**Table 7.** Metrics obtained using model stacking techniques optimized with class balancing and hyperparameter optimization strategies.

| | ROC AUC | F1 Score | Precision | Recall | Accuracy |
|---|---|---|---|---|---|
| Mortality | 0.650 | 0.351 | 0.253 | 0.574 | 0.646 |
| Length of Stay | 0.587 | 0.615 | 0.590 | 0.642 | 0.564 |

The metrics obtained were superior to those of the base models, especially for length of stay.

4.4.2. Model Validation

The results achieved in the first approximations already suggested the need to make adjustments to obtain more robust models with more conclusive metrics.

For the optimization, the poor performances of the recall and f-beta indicators were analyzed. There were imbalance problems in the data set used. This was the case for the ICU survival data (survival during hospital admission in the critical unit); from a total of 24,876 patients, the number of patients who survived (20,731) was much higher than the number of patients who died (4145).

It was necessary to consider that, in clinical contexts, it is very common for data to be unbalanced. That is, there is an unequal number of elements belonging to one class or another.

To improve the obtained results, resampling techniques and the generation of synthetic samples were carried out [36], which are recommended to compensate for the imbalance in the data set [37]—in this case, the disproportion in the survival rate.

Specifically, we obtained the best results by applying oversampling techniques, namely, the minority class synthetic oversampling technique [38], and undersampling techniques, namely, Tomek links (optimization of the Condensed Nearest Neighbors technique [39]).

In addition, a hyperparameter optimization process was carried out, specifically using Grid Search, to build and evaluate the models with different hyperparameter configurations until the most accurate model was obtained.

We calculated the Receiver Operating Characteristic (ROC) values, and the curve is displayed in Figure 5.

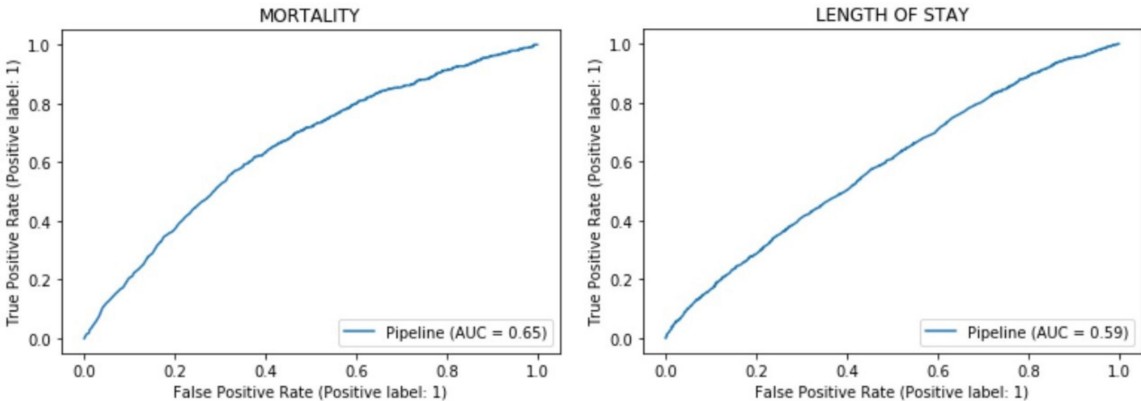

**Figure 5.** Receiver Operating Characteristic (ROC) curve for the outcome variables of mortality and length of stay obtained with the intelligent model.

The obtained metrics allowed for analysis of the model. In relation to mortality, the accuracy was reasonable, although the precision and recall were mediocre. Regarding length of stay, the accuracy was similar to that of mortality, and the recall and precision were much better. In both cases, the ROC AUC indicated that the predictions were valid, but it did not provide a definitive degree of confidence.

*4.5. Contrasting the Intelligent Model and the Scoring Model*

To assess the goodness of the intelligent model, a comparative analysis of the results with a traditional scoring system was carried out following a consolidated criterion. The purpose was to verify that an improvement would be obtained using the intelligent model, which globally outperformed the traditional model.

Regarding the scoring model, an ex post analysis of the implementation of the Philip R. Lee Institute for Health Policy Studies criteria was carried out [40].

The 24,876 ICU admission records used in the generation of the intelligent model were processed using the indicated scoring model, obtaining predictions of survival and length of stay with the metrics indicated in Table 8.

**Table 8.** Scoring model metrics for the outcome variables of mortality and length of stay obtained using the intelligent model.

|  | **F1 Score** | **Precision** | **Recall** | **Accuracy** |
|---|---|---|---|---|
| Mortality | 0.112 | 0.116 | 0.108 | 0.774 |
| Length of Stay | 0.412 | 0.549 | 0.329 | 0.566 |

When analyzing the intelligent model and the scoring model, the following results were observed:

- For mortality, the intelligent model had an accuracy of 0.646 and an F1 score of 0.351, while the scoring model had a higher accuracy of 0.774, but the F1 score was much lower. The scoring model had an extremely low recall of 0.108 compared to 0.574 for the intelligent model, so the scoring model failed to correctly identify most deaths. The precision was superior in the intelligent model.
- As for length of stay, the accuracy was similar in both models. The intelligent model had a significantly higher recall than the scoring model, indicating a better ability to identify long stays. It also had a better F1 score of 0.615 versus 0.412 for the scoring model, indicating a better balance between accuracy and recall.
- The intelligent model prevailed over the scoring model for both outcome variables, providing a better balance of metrics, which is key in a medical context.

The best performance of the intelligent model is visually shown in Figure 6.

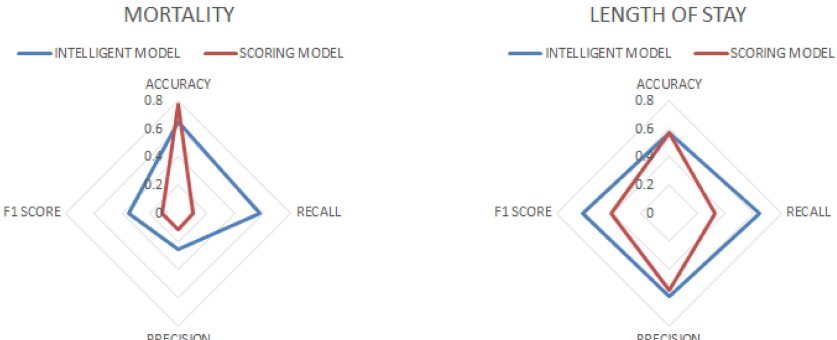

**Figure 6.** Comparison of the performance metrics of the scoring model and the intelligent model.

## 5. Discussion

The presented results show that the use of big data models and artificial intelligence algorithms allows results to be obtained that surpass traditional algorithmic systems based on the experience of clinical professionals and can be applied as scoring systems. It was also shown that using solid architectures that allow interoperability between data sources helps to improve the coherence of information and allows for a data repository large enough to generate intelligent models to be obtained. Another important aspect of integration is a system that can be maintained continuously, allowing for evolution and improvement of the models obtained. The way to ensure the repeatability of the process is the establishment of a methodology.

The applicability of the model in clinical settings suggests a significant advancement in medical decision support capabilities. This model, which uses advanced machine learning techniques to predict length of stay and survival rates in critical care units, provides healthcare professionals with a tool to improve resource management and patient care. When comparing the efficacy of our model to traditional scoring systems, it is observed to offer superior precision in predicting critical clinical outcomes. This improved precision can be crucial in optimizing hospital resources and care planning, especially in high-demand and resource-limited settings.

The assessment of metrics, although surpassing the traditional scoring model, also shows that there is considerable potential for improvement. This can be clearly seen in the

ROC metrics obtained for the estimation of mortality and length of stay (AUC 0.65 and 0.59, respectively).

During the development of the study, methodological limitations were identified due to the dataset being restricted to a single center, which made it susceptible to overfitting. To enhance the model's generalizability and reduce the risk of overfitting, specific measures were implemented, including the use of resampling techniques, the generation of synthetic samples, and hyperparameter optimization through grid search. Another limitation was the use of natural language processing techniques, which inevitably led to erroneous interpretations of some terms (e.g., abbreviations). More rigorous data collection by the center is required in order to increase the reliability of the model, especially the use of standardized nomenclature.

Another limitation identified was the use of a single set of criteria (PRLI criteria). Although this is a robust set of criteria, alternative sets of criteria could be used for cross-comparison.

Although the model showed marked improvement in several key performance metrics, it is necessary to recognize the limitations inherent in the single-center study design and the homogeneity of the data set. These factors may limit its applicability in different clinical environments. Further studies incorporating multicenter data are essential to validate and refine the predictive model, ensuring that it can work effectively in a broader spectrum of clinical environments.

## 6. Conclusions

With the large volume of data and some uncertainty about the most optimal methods for obtaining results, it is necessary to make approximations for the selection of those techniques that are most appropriate in terms of their efficacy or performance, and, based on their selection, to make the costliest adjustments and optimizations.

However, it is important to maintain the objectives proposed, not focusing on the improvement of any specific metric arbitrarily, but with awareness of its functional relevance from a clinical point of view. It is important to consider that decisions with clinical impact, whether related to patient diagnosis and treatment or patient management, should always be made by the clinicians, who can benefit from the information provided by an intelligent system while using their knowledge and experience, as well as taking responsibility for the decisions they adopt.

The methodology proves to be essential to organize and control the execution of multiple heterogeneous tasks in pursuit of a specific purpose to analyze data and make decisions, while also guaranteeing the soundness and discipline of the process and the validity of the data obtained.

From a technical point of view, future research will focus on improving the results. This improvement can be approached from two fronts. On the one hand, both the volume and the type of information the system is fed can be increased. On the other hand, other options can be explored in the application of AI that allow for the improvement of the percentage of success in the predictions. Another clear continuation of this study would be the development of an application that would allow for the exploitation of the model by applying it to the assessment of future patients in the critical care unit.

From a clinical perspective, the model can be extended to the prioritization of patients who are candidates for admission to the intensive care unit, thus facilitating the identification of patients who would benefit from early admission to the ICU [41]. It would also facilitate the management of patients admitted to the ICU to improve hemodynamic management [42,43], anticipate the complications associated with mechanical ventilation [44], and anticipate intubation [45].

For future research, it is also recommended to extend the methodology to the development of intelligent models for non-critical hospitalization, with a particular emphasis on various clinical specialties.

**Author Contributions:** Conceptualization, S.O.-T. and E.M.B.; methodology, S.O.-T. and J.-M.G.-M.; software, E.M.B.; validation, E.M.B. and S.O.-T.; formal analysis, E.M.B. and A.C.-M.; investigation, E.M.B. and J.-M.G.-M.; resources, E.M.B.; data curation, E.M.B.; writing—original draft preparation, E.M.B.; writing—review and editing, J.-M.G.-M.; visualization, A.C.-M. and J.-M.G.-M.; supervision, S.O.-T. and J.-M.G.-M.; project administration, E.M.B. All authors have read and agreed to the published version of the manuscript.

**Funding:** This research received no external funding.

**Institutional Review Board Statement:** Not applicable.

**Informed Consent Statement:** Not applicable.

**Data Availability Statement:** Due to the sensitive nature of the data, the clinical center that provided the data has not authorized public distribution, so supporting data are not available.

**Conflicts of Interest:** The authors declare no conflicts of interest. The funders had no role in the design of the study; in the collection, analyses, or interpretation of data; in the writing of the manuscript; or in the decision to publish the results.

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
