# Peer review of "An Intelligent Model and Methodology for Predicting Length of Stay and Survival in a Critical Care Hospital Unit"

_informatics, doi:10.3390/informatics11020034_

Round 1

Reviewer 1 Report (Previous Reviewer 1)

Comments and Suggestions for Authors

Authors have successfully addressed all my previous comments. 

Comments on the Quality of English Language

n/a

Author Response

Dear reviewer,

Thank you very much for the contributions and for giving us the opportunity to improve the manuscript.

Kind regards

Reviewer 2 Report (Previous Reviewer 3)

Comments and Suggestions for Authors

ORIGINAL COMMENT

GENERAL I appreciate the efforts made by authors in addressing the issues raised. However, there is a major problem in the provided review: responses are very generic and the direct reference to the new version of the manuscript (punctual indication of parts of the text removed, added or reworked) is not provided. As a consequence, it is very hard to assess if some of the comments were actually solved in the new version of the manuscript. Additionally, I still have concerns about many of the raised issues, which, in my opinion, were not treated with the necessary attention. Specifically:

RESPONSE

Thank you for the comments, and the opportunity to improve the quality of the manuscript. We have made considerable changes to the manuscript, including increasing the size of the dataset, using a single dataset for both models, simplifying the structure of methods and materials, reorganizing the information between results, discussion and conclusions, and, above all, making the section on artificial intelligence more comprehensive, rigorous and clear.

We have tried to respond to comments by indicating, whenever possible, where the manuscript has been updated.

NEW COMMENT

R.3 A punctual indication of parts of the text removed, added or reworked is again not provided in this review, so the main issue still remains. This made the review activity a lot harder to perform.

I appreciate authors effort in addressing some of the raised points, but many of them are still unanswered. Also, there are some additional concerns relevant to the new version.

ORIGINAL COMMENT

Comment 1.1):

R1. I believe that, regardless the evaluated performance of the proposed model, its use as a primary decision maker is not to be recommended. Until extensive and rigorous on-field assessments are performed, the role for the model indicated by authors should be as a support for human decisionmakers. This is an ethical issue of primary importance. I therefore suggest to re-consider a modify all statements in the text relevant to this matter, (such as, for example, <<The result of a model with these characteristics would provide security in decisions, reducing their subjectivity>>) and to clarify this aspect."

R2. Please indicate punctually the changes (parts of the text removed, parts of the text added) made to address this issue.

We agree that the AI solution should be supportive, and that the final decision should be made by the responsible clinician. We have indicated this point in the second paragraph of the conclusions.

NEW COMMENT

R3. Thank you for addressing this issue. Were the more assertive parts of the text removed as well?

ORIGINAL COMMENT

Comment 1.2):

R1. In chapter 2 (methods and materials), it is necessary to add, in the introductory section (2.1), the study context of the specific study-case. Mainly, please describe the origin and the features (basic population statistics) of used data, about which nothing at all is reported in the current version of the manuscript. Also, data dimension should be put here rather than in the discussion."

R2. I find it hard to believe that “no missing or inconsistent data were identified” across 22 years of data, dating back to 1999. Unfortunately, I see from data availability statement that authors are not allowed to disclose original data.

RESPONSE

The problems we identified stemmed primarily from the use of natural language processing. We have included that limitation in the third paragraph of the discussion. We have also expanded the volume of the dataset, to include the most recent data.

NEW COMMENT

R3. This does not address the raised issue.

ORIGINAL COMMENT

Comment 1.3): 

R1. Methods and materials: on top of describing the goals of the various phases/tasks, it is also necessary to describe how they are carried out in the presented study-case, which is currently reported (at least in part) in the results. I understand that the definition of the workflow is a primary part of the methodology, but so is the actual implementation in the addressed stud-case, whereas the results are the quantitative/qualitative data (or information in general) that is generated through the application of what was implemented (output). Additionally, the baseline reference method for patients’ assessment (which is compared to ML model) should be presented in details in this section."

R2. I can’t find in the manuscript the content matching authors’ response. Please indicate it punctually.

RESPONSE

We have updated the structure of the methodology in order to make it clearer. We have also simplified and made the comparison more balanced by unifying the data source for the comparison of the two models. 

The update of the dataset is shown in section 3.1, and we clarify the process of the methodology in the first paragraph of section 3.2.

NEW COMMENT

R3. Despite the changes, this is still a description of a general framework, with minimal reference to the actual study-case implementation on which the study is proposed.

ORIGINAL COMMENT

Comment 1.5):

R1. The features selection process is not described in detail. By reading the manuscript, what I understand is that explicative variables are those reported in table 1 (which are all binary variables), is that correct? Were explicative variables somehow pre-processed and selected? For example, was collinearity between the variables assessed?"

R2. If I understand properly, the variables selection is based on a literature reference, and no further processing is applied. Also due to the fact that this reference is dated 2009, I think that a more rigorous approach would be necessary when it comes to implementing a ML model.

RESPONSE

Indeed, we start from the Philip R. Lee Institute's criteria as the standard for carrying out the analysis. It is possible to locate other criteria, but the one selected is a solid criterion that allows us to realistically contrast both models.

The possibility of a comparative analysis between different criteria could be of interest, although it would exceed the scope of the manuscript.

NEW COMMENT

R3. While the reference proposed by authors is undoubtedly valid, I still think that a more updated and data-driven approach is necessary. Also, the first part of the inquiry was not addressed at all.

ORIGINAL COMMENT

Comment 1.6):

R1. The validation metrics considered can be consistently improved. For instance, mortality is a binary target variable, therefore I think it is recommendable to perform a ROC-based validation considering AUC as the performance metric. Also in the comparison between traditional and Ml model, basing it on the accuracy alone is poorly significant."

R2. I appreciate the efforts made by authors to include the ROC analysis. However, it is not addressed in the methodology, and its description in the results is very poor: it is unclear what model was it implemented for, and with what target outcome.  Moreover, the obtained result (AUC 0.65) indicates a low performance, discouraging model’s application. This is not addressed in the discussion.

RESPONSE

We have made a strong update of the entire artificial intelligence section. We describe the ROC AUC in section 3.2.4, and comment on the limited performance in the second paragraph of the discussion.

NEW COMMENT

R3. I appreciate authors effort in improving this section, which is now consistently more exhaustive.

ORIGINAL COMMENT

Comment 1.7): 

R1. Methods and materials: << Therefore, it is not considered within the scope of the study to technically deepen the tools used, nor to carry out exhaustive improvements and optimizations.>> even in terms of hyper-parameters tuning? I do agree that there is no need to implement a new algorithm, but at least the basic settings should be (to some extent) optimized. For instance, with a ROC-based validation it would be possible to optimize the threshold of the labelling probability resulting from the ML block, thus improving the performance, as well as setting the dimensioning of the decision trees and random forest. Or again, the classification strategy for length of stay is arbitrary, while a testing protocol could prove which one can enable a higher output performance. As a matter of fact, the performance is quite poor, especially in predicting LOS (where the reached performance is even lower than the traditional expert-based model).

R2. I don’t see any changes made to address this issue. Please point it out clearly.

RESPONSE

It is correct that the level of detail and optimization done was limited and could be improved. Therefore we have completely rebuilt the artificial intelligence part. We have extended the set of techniques to be used, and performed optimizations and improvements (mainly class balancing and hyperparameter optimization). It is detailed in point 4.4.1

NEW COMMENT

R3. As for the previous issue, I appreciate authors effort in elaborating on this aspect, which is now consistently improved.

ORIGINAL COMMENT

Comment 1.8):

R1. I don’t understand why the dataset for the traditional model is so limited. To the best of my understanding (although, as already stated in comment 3, its functioning must be explained better) the traditional model is based on a formula applied to the same data structure. Wasn’t it possible to label them with the traditional ‘manual’ method?"

R2. I can’t find the part of the manuscript where this issue is addressed. Please indicate it punctually.

RESPONSE

We have completely eliminated the use of the reduced dataset for the traditional scoring model, and we now keep the single large dataset for comparison. We consider it much more fair and balanced.

NEW COMMENT

R3. Thank you for the adjustment, which I deem have improved the validity of the analysis.

ORIGINAL COMMENT

Comment 1.9):

R1. All figures and table captions should be (as far as possible) self-explaining. The current form is way too concise."

R2. Implemented changes are too limited to make the captions self-explanatory.

RESPONSE

We have adjusted the names of the figures and tables to make them more descriptive. We have also incorporated additional tables and figures.

NEW COMMENT

R3. I find the current captions still too concise.

ORIGINAL COMMENT

Comment 1.10):

R1. A limitations section is necessary in the discussion. Among them, please also address how the use of a specific custom NLP model, thus not replicable/extendible, can impact the model development and consequent performance."

R2. The limitations section should be beneficial for the readers, clarifying what are the boundaries of applicability of the presented research. To achieve this, a much more careful assessment should be performed, addressing the punctual issues encountered and clarifying their impact on the applicability, along with some suggestions for possible solutions. The limitations section presented in the current manuscript is minimal, and seems to be put there only to respond to reviewers concerns.

RESPONSE

We have incorporated the limitations in the third paragraph of the discussion, I have included comments on how this point could be improved in the future.

NEW COMMENT

R3. The limitations are now more clearly outlined.

ORIGINAL COMMENT

2.1) The discussion section is very short and does not provide any insight on the obtained results or their applicability.

RESPONSE

We have reorganized the information and expanded the discussion section.

NEW COMMENT

R3. Despite the inclusion of a limitations section, the discussion is still insufficient, not providing any insight on the obtained results, their relevance, or their applicability.  Also, no comparison with state-of-art is provided.

NEW ADDITIONAL COMMENTS

3.1) Authors state “In this section it is presented the reason for starting the work presented in this paper, based in the existence of a data set that was created and maintained along many years

with the aim of using it in future research but without specific objective or method set.” Does this mean that the goal of the whole study is to put at use a dataset that was previously collected with no actual goal? This seems approximative from a scientific point of view.

3.2) Authors state “Although the restriction to a single center implies a more limited data set, it also increases the homogeneity and reliability of the data analysis.” I disagree with this statement. A more homogeneous data set is more likely subject to overfitting, and this reduces the reliability of any AI-based model when applied to different data.

3.3) Authors report “The process that is carried out is as follows: A first approximation is made using various machine learning techniques. Optimizations are then performed on each of these models, obtaining results that allow discerning those with the best metrics. Among these models, a stacking model is proposed to obtain the most solid model for the prediction of both outcome variables.” What do authors mean by ‘stacking model’? Is it the series of different computational algorithms, the parallel, or else? This requires elaboration to state that the final prediction could benefit from this approach.

Additionally, this disagrees with the previous statement “It is not necessary to obtain the optimal predictive model, but rather to verify an improvement using machine learning models in relation to score systems. Therefore, it is not considered within the scope of the study to technically fine-tune the tools used, or to carry out exhaustive improvements and optimizations.”

Author Response

Dear Reviewer,

First of all, we would like to explain that the editor requested us to make changes and send the result as a new submission, and we wrongly understood that we don't have to make the document with a detailed answer for every comment. Sorry for having make you to apply unnecessary efforts.

And, of course, thank you for your comments. We have carefully reviewed all the points raised, and have made the adjustments and incorporated clarifications in this regard.

We hope that these changes will bring the manuscript to the quality required for publication.

Thank you very much for your help to improve the quality of the work.

Kind regards

Reviewer 3 Report (Previous Reviewer 2)

Comments and Suggestions for Authors

The authors have adequately responded to my concerns. Their explanations are satisfactory, enhancing the manuscript's overall quality.

Author Response

Dear reviewer,

Thank you very much for the contributions and for giving us the opportunity to improve the manuscript.

Kind regards

Round 2

Reviewer 2 Report (Previous Reviewer 3)

Comments and Suggestions for Authors

Author Response

Dear reviewer,

Thank you very much for the review, the comments provided and the opportunity to improve the work. Please see the attachment with the description of the changes made, as well as the corresponding clarifications.

Best regards

This manuscript is a resubmission of an earlier submission. The following is a list of the peer review reports and author responses from that submission.

Round 1

Reviewer 1 Report

Comments and Suggestions for Authors

Major points

1. (Page 2) It would be great if you could explain what "the evolution of patients" means. Not every author may be familiar with this term. Please elaborate on this further.

2. What is the existing literature on this? On page 6, you wrote, “Previous studies have been carried out for the authors using these criteria with sim[1]ilar objectives although using traditional calculation techniques and obtaining results that ants to be improved in this study [21].” Is there only one study on the traditional approach? You should have included more in the literature review, especially if one of your objectives is to compare your results with the traditional approach.

3. “Although the numerical results obtained are a good approximation, in order to have a comparable result between the two techniques, the best solution is to use the same data set in both cases.” If so, I don’t think it is necessary to include the results using different datasets.

4. What are the limitations of your study?

Minor

1. “Artificial intelligence is a powerful tool that makes it possible to maximize the massive use of data and obtain information that would be difficult to obtain through traditional analysis [6]. Currently, machine learning techniques are already being applied, especially for forecasting the appearance of specific diseases, as indicated by Shamout [7].” This part is a bit repetitive as you mentioned this already in the first paragraph.

2. As I am not super familiar with the approach that you took in this study, can you please explain what the hit rate is+? How can you determine whether one approach is better than the other with hit rates? Please explain.

Comments on the Quality of English Language

I think this paper would benefit from major editing services.

Author Response

Dear reviewer,

We have carefully checked your instructions and have made the appropriate changes accordingly. Please see the attachment.

Thank you for your contribution to improving the quality of the paper, and for giving us the opportunity to do better.

Sincerely

The authors

Reviewer 2 Report

Comments and Suggestions for Authors

I am thankful for the for the opportunity to review the article titled "Intelligent Model and Methodology for Predicting Length of Stay and Survival in a Critical Care Hospital Unit”.

This paper outlines the design and methodology for developing a prediciton model using artificial intelligence to predict patient length of stay and survival rates in critical care units. Results indicate that the use of Big Data models and artificial intelligence algorithms outperforms traditional clinician-based scoring systems.

I have several concerns regarding the presentation of the study:

1)      First the Mtehod section should clearly state:
Acquisition of date in more Deatail (what data is used is it available in full at what timepoint of the stay etc...)
Outcome variables
Predictor variables
Handling of missing data
Describe in detail witch methods are used for prediction and how these are tuned/selected
The methods section reads like a a priori study plan and not like a description of methods used in an already done study.

2)      Results start with a description of the data acquisition  => should be part of the method section

3)      Why is the though process behind picking a grouping method for one of the outcome parameters described in unnecessary detail? Again, this should be part of the method section

4)      Model assessment. Is there a reasoning for not using ROC for the binary model and giving typical numbers like true negative false negative etc.?

5)      “To improve the results obtained, resampling techniques and the generation of syn-thetic samples are carried out”

These techniques should be described in detail.

6)      The process of data splitting and its reasoning should be described.

7)      Discussion starts with: “In order to assess the goodness of the intelligent model, a comparative analysis of the results with a traditional scoring system has been carried out following a consolidated criterion. The purpose is to verify that an improvement is obtained in the intelligent model, which outperforms the traditional model.”

This should be described in the method section and the results in the results I don’t understand why this is part of the disciussion?

The discribtion of the models in the discussion again consists of elements that should have been described in the method section and in the results. These results should then be discussed in the discussion.

8)      Why or why not were the models validated with other data sets (several ICU data sets exist and are freely available)

Comments on the Quality of English Language

Several grammatical and spelling errors. Prof. proofreading recommended.

Author Response

(The authors gave the same response as above.)

Reviewer 3 Report

Comments and Suggestions for Authors

The proposed manuscript “Intelligent Model and Methodology for Predicting Length of Stay and Survival in a Critical Care Hospital Unit” presents a framework for the project and implementation of a machine-learning based model that uses hospital data in order to predict the survival likelihood and the length of stay of ICU patients. While I strongly appreciate the attention that authors devoted towards the schematization of the overall workflow, and the clarity of its description, I think there are several issues that should be addressed before considering the manuscript viable for publication.

MAJOR:

1) I believe that, regardless the evaluated performance of the proposed model, its use as a primary decision maker is not to be recommended. Until extensive and rigorous on-field assessments are performed, the role for the model indicated by authors should be as a support for human decision-makers. This is an ethical issue of primary importance. I therefore suggest to re-consider a modify all statements in the text relevant to this matter, (such as, for example, <<The result of a model with these characteristics would provide security in decisions, reducing their subjectivity>>) and to clarify this aspect.

2) In chapter 2 (methods and materials), it is necessary to add, in the introductory section (2.1), the study context of the specific study-case. Mainly, please describe the origin and the features (basic population statistics) of used data, about which nothing at all is reported in the current version of the manuscript. Also, data dimension should be put here rather than in the discussion.

3) Methods and materials: on top of describing the goals of the various phases/tasks, it is also necessary to describe how they are carried out in the presented study-case, which is currently reported (at least in part) in the results. I understand that the definition of the workflow is a primary part of the methodology, but so is the actual implementation in the addressed stud-case, whereas the results are the quantitative/qualitative data (or information in general) that is generated through the application of what was implemented (output). Additionally, the baseline reference method for patients’ assessment (which is compared to ML model) should be presented in details in this section.

4) In general, there are some re-displacements to be performed across section (see minor comments for details).

5) The features selection process is not described in detail. By reading the manuscript, what I understand is that explicative variables are those reported in table 1 (which are all binary variables), is that correct? Were explicative variables somehow pre-processed and selected? For example, was collinearity between the variables assessed?

6) The validation metrics considered can be consistently improved. For instance, mortality is a binary target variable, therefore I think it is recommendable to perform a ROC-based validation considering AUC as the performance metric. Also in the comparison between traditional and Ml model, basing it on the accuracy alone is poorly significant.

7) Methods and materials: << Therefore, it is not considered within the scope of the study to technically deepen the tools used, nor to carry out exhaustive improvements and optimizations.>> even in terms of hyper-parameters tuning? I do agree that there is no need to implement a new algorithm, but at least the basic settings should be (to some extent) optimized. For instance, with a ROC-based validation it would be possible to optimize the threshold of the labelling probability resulting from the ML block, thus improving the performance, as well as setting the dimensioning of the decision trees and random forest. Or again, the classification strategy for length of stay is arbitrary, while a testing protocol could prove which one can enable a higher output performance. As a matter of fact, the performance is quite poor, especially in predicting LOS (where the reached performance is even lower than the traditional expert-based model).

8) I don’t understand why the dataset for the traditional model is so limited. To the best of my understanding (although, as already stated in comment 3, its functioning must be explained better) the traditional model is based on a formula applied to the same data structure. Wasn’t it possible to label them with the traditional ‘manual’ method?

9) All figures and table captions should be (as far as possible) self-explaining. The current form is way too concise.

10) A limitations section is necessary in the discussion. Among them, please also address how the use of a specific custom NLP model, thus not replicable/extendible, can impact the model development and consequent performance.

11) There is need for an extensive language editing (some examples in comment XX).

MINOR:

12) Abstract <<The model is generated using artificial intelligence>>: it seems that AI is used in the development process rather than being the computational engine of the model, I would rephrase in <<The generated model relies on artificial intelligence>>

13) Introduction: <<the evolution of patients>>, I suggest saying ‘the evolution of patient’s clinical condition’

14) Introduction: <<However, the greatest weakness of these systems is also found in the power of the information obtained>> in light of what is reported next, I think authors might want to refer to “interpretability of the information” rather than “power”, or at least “meaningfulness”.

15) Introduction: << Previous analyses of predictive models indicate that the results obtained from predictions using machine learning techniques are not superior to linear regression [12]. However, reviews of the literature indicate that the median sample size on which we work is 1,250 records [13]. The use of a sample with a larger order of magnitude is expected to provide results that justify the use of machine learning to make predictions, as has been demonstrated in specific predictions concerning critically ill patients with larger volumetric [14]>>. Do authors mean that the equivalent performance of ML and LR stands for limited datasets only? If so, this should be clarified.

16) Chapter 2 (methods and materials): although it is easy to match sections and sub-sections to phases and tasks of the workflow, due to their limited number, I still suggest matching the titles in the text with labels in the schema.

17) Methods and materials: I guess << 2.5. Maching Learning>> is meant to be “2.5 Machine Learning”.

18) Results table 1: please graphically re-work it so that the classes in the left column can be matched to elements in the right column. Same for table 2 and 4.

19) In figure 2, I suggest swapping data lake and data warehouse, as (if I understood correctly the workflow), data coming from the interoperability block flows into the data lake, and a data warehouse is then structured from it, with this second being the basis for the following model generation.

20) Table 3: I suggest moving after the following paragraph, so that the reason to exclude strategy D is explained before the table itself.

21) Results: << Accuracy is only good applying random forest for mortality, with random forest also providing the least bad accuracy.>> This sentence is not clear. The accuracy of random forest in predicting mortality (0.83) is the same as that of logistic regression.

22) The output results presented in section 4.3 should be moved to the results chapter, and then critically analyzed in the discussion.

23) Discussion: << The purpose is to verify that an improvement is obtained in the intelligent model, which outperforms the traditional model.>> This is not completely true, as the performance in predicting LOS is lower in ML model compared to the traditional one.

LANGUAGE:

24) A consistent language editing is necessary, in terms of grammar, lexicon, and scientific writing. I suggest checking the form/language of the whole text, and of the following sentences in particular:

<<Among the possibilities of applying technology in the field of health are the integration of systems, Big Data and artificial intelligence>>

<<makes possible to overcome the barriers and limits of analysis carried out in a traditional analysis>>

<<A context where it useful to have estimations>>

<<being in practice unfeasible to know with a reasonable effort the reason why a certain prediction is made [9], because it is based on the analysis of multiple parameters of tens or hundreds of thousands of previous registrations.>>

<< Contrast the goodness of the results obtained with respect to a traditional scoring system>> use “compare” instead of “contrast” (same in the next sentence)

<< and, if necessary, the creation of new models in the area if necessary>>

<< One of the main points to develop the model is create an architecture>>

<< The following factors are selected are selected for assessment>>

<< The processes of obtaining the models are run several times by randomizing the order of the data and testing the sample percentage for training and validation>>

<< Accuracy is only good applying random forest for mortality, with random forest also providing the least bad accuracy.>>

Comments on the Quality of English Language

24) A consistent language editing is necessary, in terms of grammar, lexicon, and scientific writing. I suggest checking the form/language of the whole text, and of the following sentences in particular:

<<Among the possibilities of applying technology in the field of health are the integration of systems, Big Data and artificial intelligence>>

<<makes possible to overcome the barriers and limits of analysis carried out in a traditional analysis>>

<<A context where it useful to have estimations>>

<<being in practice unfeasible to know with a reasonable effort the reason why a certain prediction is made [9], because it is based on the analysis of multiple parameters of tens or hundreds of thousands of previous registrations.>>

<< Contrast the goodness of the results obtained with respect to a traditional scoring system>> use “compare” instead of “contrast” (same in the next sentence)

<< and, if necessary, the creation of new models in the area if necessary>>

<< One of the main points to develop the model is create an architecture>>

<< The following factors are selected are selected for assessment>>

<< The processes of obtaining the models are run several times by randomizing the order of the data and testing the sample percentage for training and validation>>

<< Accuracy is only good applying random forest for mortality, with random forest also providing the least bad accuracy.>>

Author Response

(The authors gave the same response as above.)

Round 2

Reviewer 1 Report

Comments and Suggestions for Authors

Thank you for your revision. However, there is one thing I am unsure about in the revised version. Despite the authors stating, 'We have incorporated the clarification in point 4.3,' I couldn't find Section 4.3 in the current version of the paper. Please specify.

Comments on the Quality of English Language

 Extensive editing of English language required.

Author Response

LANGUAGE English very difficult to understand/incomprehensible

In order to improve the quality of the language, a complete revision of the writing has been made using the editorial services of English language editing.

MAJOR POINTS

Thank you for your revision. However, there is one thing I am unsure about in the revised version. Despite the authors stating, 'We have incorporated the clarification in point 4.3,' I couldn't find Section 4.3 in the current version of the paper. Please specify.

In the revised version, we only work with the extended version of the dataset, in order to be able to perform a fairer and more balanced analysis and comparison, therefore this clarification does not appear in any section.

MINOR POINTS

No comments

Reviewer 3 Report

Comments and Suggestions for Authors

I appreciate the efforts made by authors in addressing the issues raised. However, there is a major problem in the provided review: responses are very generic and the direct reference to the new version of the manuscript (punctual indication of parts of the text removed, added or reworked) is not provided. As a consequence, it is very hard to assess if some of the comments were actually solved in the new version of the manuscript. Additionally, I still have concerns about many of the raised issues, which, in my opinion, were not treated with the necessary attention. Specifically:

PREVIOUS COMMENTS:

Comment 1.1):

I believe that, regardless the evaluated performance of the proposed model, its use as a primary decision maker is not to be recommended. Until extensive and rigorous on-field assessments are performed, the role for the model indicated by authors should be as a support for human decision-makers. This is an ethical issue of primary importance. I therefore suggest to re-consider a modify all statements in the text relevant to this matter, (such as, for example, <<The result of a model with these characteristics would provide security in decisions, reducing their subjectivity>>) and to clarify this aspect.

In fact, decision making with clinical impact should always be the responsibility of the clinician, both from an ethical and legal point of view. For this reason, we have included a paragraph closing the conclusions indicating and developing this point.

Please indicate punctually the changes (parts of the text removed, parts of the text added) made to address this issue.

Comment 1.2):

In chapter 2 (methods and materials), it is necessary to add, in the introductory section (2.1), the study context of the specific study-case. Mainly, please describe the origin and the features (basic population statistics) of used data, about which nothing at all is reported in the current version of the manuscript. Also, data dimension should be put here rather than in the discussion.

We have incorporated the context of the case study as well as the dimension of the data in the introduction of methods and materials section.

I find it hard to believe that “no missing or inconsistent data were identified” across 22 years of data, dating back to 1999. Unfortunately, I see from data availability statement that authors are not allowed to disclose original data.

Comment 1.3):

Methods and materials: on top of describing the goals of the various phases/tasks, it is also necessary to describe how they are carried out in the presented study-case, which is currently reported (at least in part) in the results. I understand that the definition of the workflow is a primary part of the methodology, but so is the actual implementation in the addressed stud-case, whereas the results are the quantitative/qualitative data (or information in general) that is generated through the application of what was implemented (output). Additionally, the baseline reference method for patients’ assessment (which is compared to ML model) should be presented in details in this section.

Effectively, in the methods section we were giving more priority to the methodology of obtaining the model than to the justification of its application.

We have gone deeper into the points indicated and have incorporated them into the Materials and Methods section. We also specify the reference criteria for the comparison of the two methods.

I can’t find in the manuscript the content matching authors’ response. Please indicate it punctually.

Comment 1.4):

In general, there are some re-displacements to be performed across section (see minor comments for details).

We have reviewed the minor points and made the indicated adjustments.

Checked.

Comment 1.5):

The features selection process is not described in detail. By reading the manuscript, what I understand is that explicative variables are those reported in table 1 (which are all binary variables), is that correct? Were explicative variables somehow pre-processed and selected? For example, was collinearity between the variables assessed?

For the selection of characteristics, we use the criteria of the Philip R. Lee Institute, as a contrasted standard to perform the analyses. These are indeed those indicated in Table I; we have emphasized the clarification. We did not make any selection within the variables, nor did we study collinearity, which is likely to occur considering that some of the criteria are above an age threshold. We consider the idea to be very interesting, and it could be worthwhile in subsequent studies on ICU patients.

If I understand properly, the variables selection is based on a literature reference, and no further processing is applied. Also due to the fact that this reference is dated 2009, I think that a more rigorous approach would be necessary when it comes to implementing a ML model.

Comment 1.6):

The validation metrics considered can be consistently improved. For instance, mortality is a binary target variable, therefore I think it is recommendable to perform a ROC-based validation considering AUC as the performance metric. Also in the comparison between traditional and Ml model, basing it on the accuracy alone is poorly significant.

We have included the ROC curve to help complete the assessment of the rest of the parameters.

I appreciate the efforts made by authors to include the ROC analysis. However, it is not addressed in the methodology, and its description in the results is very poor: it is unclear what model was it implemented for, and with what target outcome.  Moreover, the obtained result (AUC 0.65) indicates a low performance, discouraging model’s application. This is not addressed in the discussion.

Comment 1.7):

Methods and materials: << Therefore, it is not considered within the scope of the study to technically deepen the tools used, nor to carry out exhaustive improvements and optimizations.>> even in terms of hyper-parameters tuning? I do agree that there is no need to implement a new algorithm, but at least the basic settings should be (to some extent) optimized. For instance, with a ROC-based validation it would be possible to optimize the threshold of the labelling probability resulting from the ML block, thus improving the performance, as well as setting the dimensioning of the decision trees and random forest. Or again, the classification strategy for length of stay is arbitrary, while a testing protocol could prove which one can enable a higher output performance. As a matter of fact, the performance is quite poor, especially in predicting LOS (where the reached performance is even lower than the traditional expert-based model).

We have increased the level of detail and added the ROC curve to increase the information provided.

In fact, the prediction of hospital stay is not remarkable; it is in the mortality predictor where there are notable improvements.

I don’t see any changes made to address this issue. Please point it out clearly.

Comment 1.8):

I don’t understand why the dataset for the traditional model is so limited. To the best of my understanding (although, as already stated in comment 3, its functioning must be explained better) the traditional model is based on a formula applied to the same data structure. Wasn’t it possible to label them with the traditional ‘manual’ method?

Although the restriction to a single center implies a more limited data set, it also increases the homogeneity and reliability of the data analysis. This is due to the fact that the different clinical centers are heterogeneous, as are the patients admitted to them, and each center has different casuistry and different survival and length of stay rates. We have incorporated this clarification in the section on materials and methods. Manual labeling was carried out on the dataset reduced directly by clinicians within the hospital operations, so it is not feasible to cover it for the complete dataset (with which the model is generated). Anyway, as you indicate in comment 3, we have deepened the description.

I can’t find the part of the manuscript where this issue is addressed. Please indicate it punctually.

Comment 1.9):

All figures and table captions should be (as far as possible) self-explaining. The current form is way too concise.

We have made the figures and table captions more detailed and descriptive.

Implemented changes are too limited to make the captions self-explanatory.

Comment 1.10):

A limitations section is necessary in the discussion. Among them, please also address how the use of a specific custom NLP model, thus not replicable/extendible, can impact the model development and consequent performance.

The limitations were not mentioned, so we have included them in the discussion section, and included the use of custom NLP model

The limitations section should be beneficial for the readers, clarifying what are the boundaries of applicability of the presented research. To achieve this, a much more careful assessment should be performed, addressing the punctual issues encountered and clarifying their impact on the applicability, along with some suggestions for possible solutions. The limitations section presented in the current manuscript is minimal, and seems to be put there only to respond to reviewers concerns.

MINOR COMMENTS: Checked.

NEW ADDITIONAL COMMENTS:

2.1) The discussion section is very short and does not provide any insight on the obtained results or their applicability.

Comments on the Quality of English Language

English improved.

Author Response

GENERAL I appreciate the efforts made by authors in addressing the issues raised. However, there is a major problem in the provided review: responses are very generic and the direct reference to the new version of the manuscript (punctual indication of parts of the text removed, added or reworked) is not provided. As a consequence, it is very hard to assess if some of the comments were actually solved in the new version of the manuscript. Additionally, I still have concerns about many of the raised issues, which, in my opinion, were not treated with the necessary attention. Specifically:

Thank you for the comments, and the opportunity to improve the quality of the manuscript. We have made considerable changes to the manuscript, including increasing the size of the dataset, using a single dataset for both models, simplifying the structure of methods and materials, reorganizing the information between results, discussion and conclusions, and, above all, making the section on artificial intelligence more comprehensive, rigorous and clear.
We have tried to respond to comments by indicating, whenever possible, where the manuscript has been updated.

LANGUAGE        Extensive editing of English language required. Several grammatical and spelling errors.

In order to improve the quality of the language, a complete revision of the writing has been made using the editorial services of English language editing.

Comment 1.1):

I believe that, regardless the evaluated performance of the proposed model, its use as a primary decision maker is not to be recommended. Until extensive and rigorous on-field assessments are performed, the role for the model indicated by authors should be as a support for human decision-makers. This is an ethical issue of primary importance. I therefore suggest to re-consider a modify all statements in the text relevant to this matter, (such as, for example, <<The result of a model with these characteristics would provide security in decisions, reducing their subjectivity>>) and to clarify this aspect."

Please indicate punctually the changes (parts of the text removed, parts of the text added) made to address this issue.

We agree that the AI solution should be supportive, and that the final decision should be made by the responsible clinician. We have indicated this point in the second paragraph of the conclusions.

Comment 1.2):

In chapter 2 (methods and materials), it is necessary to add, in the introductory section (2.1), the study context of the specific study-case. Mainly, please describe the origin and the features (basic population statistics) of used data, about which nothing at all is reported in the current version of the manuscript. Also, data dimension should be put here rather than in the discussion."

I find it hard to believe that “no missing or inconsistent data were identified” across 22 years of data, dating back to 1999. Unfortunately, I see from data availability statement that authors are not allowed to disclose original data.

The problems we identified stemmed primarily from the use of natural language processing. We have included that limitation in the third paragraph of the discussion. We have also expanded the volume of the dataset, to include the most recent data.

Comment 1.3):

Methods and materials: on top of describing the goals of the various phases/tasks, it is also necessary to describe how they are carried out in the presented study-case, which is currently reported (at least in part) in the results. I understand that the definition of the workflow is a primary part of the methodology, but so is the actual implementation in the addressed stud-case, whereas the results are the quantitative/qualitative data (or information in general) that is generated through the application of what was implemented (output). Additionally, the baseline reference method for patients’ assessment (which is compared to ML model) should be presented in details in this section."

I can’t find in the manuscript the content matching authors’ response. Please indicate it punctually.

We have updated the structure of the methodology in order to make it clearer. We have also simplified and made the comparison more balanced by unifying the data source for the comparison of the two models.

The update of the dataset is shown in section 3.1, and we clarify the process of the methodology in the first paragraph of section 3.2.

Comment 1.4):

In general, there are some re-displacements to be performed across section (see minor comments for details)."

Checked.

OK.

Comment 1.5):

The features selection process is not described in detail. By reading the manuscript, what I understand is that explicative variables are those reported in table 1 (which are all binary variables), is that correct? Were explicative variables somehow pre-processed and selected? For example, was collinearity between the variables assessed?"

If I understand properly, the variables selection is based on a literature reference, and no further processing is applied. Also due to the fact that this reference is dated 2009, I think that a more rigorous approach would be necessary when it comes to implementing a ML model.

Indeed, we start from the Philip R. Lee Institute's criteria as the standard for carrying out the analysis. It is possible to locate other criteria, but the one selected is a solid criterion that allows us to realistically contrast both models.

The possibility of a comparative analysis between different criteria could be of interest, although it would exceed the scope of the manuscript.

Comment 1.6):

The validation metrics considered can be consistently improved. For instance, mortality is a binary target variable, therefore I think it is recommendable to perform a ROC-based validation considering AUC as the performance metric. Also in the comparison between traditional and Ml model, basing it on the accuracy alone is poorly significant."

I appreciate the efforts made by authors to include the ROC analysis. However, it is not addressed in the methodology, and its description in the results is very poor: it is unclear what model was it implemented for, and with what target outcome.  Moreover, the obtained result (AUC 0.65) indicates a low performance, discouraging model’s application. This is not addressed in the discussion.

We have made a strong update of the entire artificial intelligence section. We describe the ROC AUC in section 3.2.4, and comment on the limited performance in the second paragraph of the discussion.

Comment 1.7):

Methods and materials: << Therefore, it is not considered within the scope of the study to technically deepen the tools used, nor to carry out exhaustive improvements and optimizations.>> even in terms of hyper-parameters tuning? I do agree that there is no need to implement a new algorithm, but at least the basic settings should be (to some extent) optimized. For instance, with a ROC-based validation it would be possible to optimize the threshold of the labelling probability resulting from the ML block, thus improving the performance, as well as setting the dimensioning of the decision trees and random forest. Or again, the classification strategy for length of stay is arbitrary, while a testing protocol could prove which one can enable a higher output performance. As a matter of fact, the performance is quite poor, especially in predicting LOS (where the reached performance is even lower than the traditional expert-based model).

I don’t see any changes made to address this issue. Please point it out clearly.

It is correct that the level of detail and optimization done was limited and could be improved. Therefore we have completely rebuilt the artificial intelligence part. We have extended the set of techniques to be used, and performed optimizations and improvements (mainly class balancing and hyperparameter optimization). It is detailed in point 4.4.1

Comment 1.8):

I don’t understand why the dataset for the traditional model is so limited. To the best of my understanding (although, as already stated in comment 3, its functioning must be explained better) the traditional model is based on a formula applied to the same data structure. Wasn’t it possible to label them with the traditional ‘manual’ method?"

I can’t find the part of the manuscript where this issue is addressed. Please indicate it punctually.

We have completely eliminated the use of the reduced dataset for the traditional scoring model, and we now keep the single large dataset for comparison. We consider it much more fair and balanced.

Comment 1.9):

All figures and table captions should be (as far as possible) self-explaining. The current form is way too concise."

Implemented changes are too limited to make the captions self-explanatory.

We have adjusted the names of the figures and tables to make them more descriptive. We have also incorporated additional tables and figures.

Comment 1.10):

A limitations section is necessary in the discussion. Among them, please also address how the use of a specific custom NLP model, thus not replicable/extendible, can impact the model development and consequent performance."

The limitations section should be beneficial for the readers, clarifying what are the boundaries of applicability of the presented research. To achieve this, a much more careful assessment should be performed, addressing the punctual issues encountered and clarifying their impact on the applicability, along with some suggestions for possible solutions. The limitations section presented in the current manuscript is minimal, and seems to be put there only to respond to reviewers concerns.

We have incorporated the limitations in the third paragraph of the discussion, I have included comments on how this point could be improved in the future.

2.1) The discussion section is very short and does not provide any insight on the obtained results or their applicability.

We have reorganized the information and expanded the discussion section.

LANGUAGE

English improved.

OK.